# Physical activity, immune function and inflammation in kidney patients (the PINK study): a feasibility trial protocol

Patrick James Highton,[1,2] Jill Neale,[2] Thomas J Wilkinson,[2] Nicolette C Bishop,[1,2] Alice C Smith[1,2]

► Prepublication history and additional material are available. To view these files please visit the journal online (http://dx.doi.org/ 10.1136/bmjopen-2016-014713).

[1]School of Sport, Exercise and Health Sciences, Loughborough University, Loughborough, UK
[2]Department of Infection, Immunity and Inflammation, University of Leicester, Leicester, UK

**Correspondence to**
Patrick James Highton;
p.highton@lboro.ac.uk

## ABSTRACT

**Introduction** Patients with chronic kidney disease (CKD) display increased infection-related mortality and elevated cardiovascular risk only partly attributed to traditional risk factors. Patients with CKD also exhibit a pro-inflammatory environment and impaired immune function. Aerobic exercise has the potential to positively impact these detriments, but is under-researched in this patient population. This feasibility study will investigate the effects of acute aerobic exercise on inflammation and immune function in patients with CKD to inform the design of larger studies intended to ultimately influence current exercise recommendations.

**Methods and analysis** Patients with CKD, including renal transplant recipients, will visit the laboratory on two occasions, both preceded by appropriate exercise, alcohol and caffeine restrictions. On visit 1, baseline assessments will be completed, comprising anthropometrics, body composition, cardiovascular function and fatigue and leisure time exercise questionnaires. Participants will then undertake an incremental shuttle walk test to estimate predicted peak $O_2$ consumption ($VO_2$peak). On visit 2, participants will complete a 20 min shuttle walk at a constant speed to achieve 85% estimated $VO_2$peak. Blood and saliva samples will be taken before, immediately after and 1 hour after this exercise bout. Muscle $O_2$ saturation will be monitored throughout exercise and recovery. Age and sex-matched non-CKD 'healthy control' participants will complete an identical protocol. Blood and saliva samples will be analysed for markers of inflammation and immune function, using cytometric bead array and flow cytometry techniques. Appropriate statistical tests will be used to analyse the data.

**Ethics and dissemination** A favourable opinion was granted by the East Midlands-Derby Research Ethics Committee on 18 September 2015 (ref 15/EM/0391), and the study was approved and sponsored by University Hospitals of Leicester Research and Innovation (ref 11444). The study was registered with ISRCTN (ref 38935454). The results will be presented at relevant conferences, and it is anticipated that the reports will be published in appropriate journals in 2018.

## INTRODUCTION

Patients with all stages of chronic kidney disease (CKD) have elevated cardiovascular disease (CVD) risk that cannot be fully

## Strengths and limitations of this study

► Inclusion of control group matched for age and sex.
► A variety of outcome measures to inform future study design.
► Pragmatic study design which will inform future exercise recommendations for patients with CKD.
► No non-exercise control visit.

explained by traditional risk factors.[1] CVD is the most common cause of death among patients with CKD (23% in 2013), followed by infection (19% in 2013).[2]

Patients with CKD are often sarcopenic and obese,[3–6] and further deterioration in these characteristics is often observed after renal transplantation.[7–9] Patients with CKD and renal transplant recipients (RTRs) also frequently suffer from chronic systemic inflammation,[10 11] which can worsen cachexia and increase cardiovascular risk in CKD.[12 13] Patients with CKD and RTRs also display impaired cellular immune function,[14–16] which may explain why infection is the second-leading cause of death in this population.[2] This effect (ie, impaired immune function) is further compounded in RTRs by the immunosuppressive drug regime. Impaired immune function may further exacerbate inflammation due to alterations in circulating immune cell subsets[17] which could serve to worsen body composition and increase cardiovascular risk. Finally, patients with CKD display elevated levels of circulating prothrombotic microparticles (MPs),[18] which may worsen CVD risk and/or burden.[19]

Exercise has the potential to benefit and improve many of the aforementioned inter-related morbidities. In the general population, exercise can positively impact on weight gain,[20] muscle wasting,[21] physical capacity[22] and fatigue.[23] Furthermore, moderate aerobic exercise can also modify systemic inflammation,[24 25] bolster immune

function via alterations in circulating immune cell populations and activity and reduce circulating MP levels.[26–29]

In the CKD population, similar positive effects of exercise on body composition, physical function and quality of life have been demonstrated[30–33]; however, the research on inflammation and immune function in this population is limited. Previous research has shown that 30 min of moderate intensity walking exercise elicited a normal pattern of leucocyte and monocyte activation, while enhancing granulocyte function and promoting an anti-inflammatory environment (increased IL-10 release) in patients with pre-dialysis CKD.[34] However, in a similar population, an exhaustive bout of cycling exercise elicited a shift towards the more proinflammatory CD16+ monocyte, potentially favouring inflammation.[35] This disparity may be due to the exercise intensities investigated—moderate intensity exercise is more beneficial in strengthening immune function and preventing infection[26] and is promoted in general exercise guidelines. As such, investigating the effects of moderate intensity exercise in this patient population is more pragmatic and will help to guide future exercise recommendations, which are lacking in this patient population.

Therefore, this feasibility study aims to investigate the effects of acute aerobic walking exercise on the inflammation and immune function in patients with CKD. This will create the basis from which a larger trial can be conducted, including data for a power calculation, which will ultimately help to inform exercise recommendations in this population which are currently lacking, particularly with regards to immune function and inflammation.[36] However, a preliminary power calculation (GPower V.3.1) based on the findings of Viana et al[34] concerning total lymphocyte count (based on a statistical power of 0.80 and an alpha level of 0.05) suggests that 15 participants per group (eg, predialysis CKD, RTRs and healthy controls) will be sufficient.

## METHODS AND ANALYSIS
### Outcomes to be measured
This study will investigate the effect of 20 min of moderate intensity walking exercise on immune cell subsets and inflammatory markers. Participants will be grouped based on their status (ie, 'patient' or 'healthy control'). Both groups will complete an identical acute, cross-sectional study protocol necessitating two study visits, as explained below.

### Participant recruitment
All patients with CKD attending outpatient clinics within the University Hospitals of Leicester renal network will be screened for eligibility by their consultant nephrologist prior to recruitment, and approached during their routine outpatient appointments. The inclusion and exclusion criteria for patients are summarised in table 1. Healthy age-matched and sex-matched controls with no known chronic disease will be recruited from the local community. Those who do not believe themselves to suffer from any significant chronic disease will be eligible to participate. The 'broad' inclusion and exclusion criteria allow the incorporation of several different groups (ie, patients with non-dialysis CKD, with dialysis, RTRs and healthy controls), and thus will allow comparison between these groups. Similarly, the wide range of CKD stage will allow the influence of remaining renal function on the measured markers to be investigated.

### Trial design and timeline
This is a non-randomised, controlled, feasibility study. Participants will complete two study visits as described below.

### Visit 1
Participants will arrive at the laboratory in the morning, unfasted but having consumed no caffeine or alcohol and completed no strenuous exercise for 24 hours. Participants will complete questionnaires about time spent in leisure activity and their perception of fatigue, and assessments of anthropometry, body composition and cardiovascular condition. Participants will then undertake the Incremental Shuttle Walk Test (ISWT), followed by the Endurance Shuttle Walk Test (ESWT) as explained below. If the participant cannot complete the full 20 min duration of the ESWT, they will be withdrawn from the study.

### Visit 2
Participants will arrive at the laboratory in the morning (8–10 am, to minimise the influence of diurnal variation

| **Table 1** Inclusion and exclusion criteria for patients | |
|---|---|
| **Inclusion criteria** | **Exclusion criteria** |
| ► Established chronic kidney disease (all stages will be eligible including those with an established kidney transplant and those receiving dialysis treatment) | ► Age under 18 years<br>► Pregnancy<br>► Received kidney transplant less than 6 months prior to study entry<br>► Any element of study assessment protocol considered by principle care provider to be contraindicated due to physical impairment, comorbidity or any other reason<br>► Inability to give informed consent for any reason<br>► Visual or hearing impairment or insufficient command of English to give informed consent or comply with the assessment protocol |

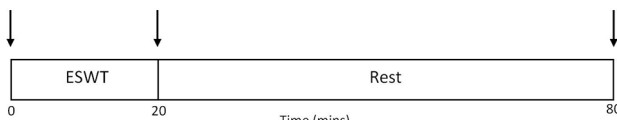

**Figure 1** Visit 2 exercise and sample collection protocol. ↓, venous blood and saliva collection. ESWT, Endurance Shuttle Walk Test.

on immune parameters), following the same standardisation procedure as in visit 1. After resting for 10 min, blood and saliva samples will be collected. The ESWT will then be completed at the same speed as in visit 1, lasting for 20 min. Within 5 min of exercise cessation, another blood and saliva sample will be collected. The participant will then rest for 1 hour, after which a final blood and saliva samples will be collected. This protocol is summarised in a schematic in figure 1, and the outcome measures are explained in greater detail below.

The time delay between recruitment, visit 1 and visit 2 will be kept to a minimum to prevent deconditioning and minimise dropout rates. For dialysis patients, both study visits will be completed on a non-dialysis day that is not after their 'long break' (ie, weekend or two consecutive days without dialysis based on their regular shift pattern) to minimise the effect of fluid overload.

### Physical performance
Endurance capacity will be assessed using the ISWT[37] and ESWT.[38] In the ISWT, the participant walks a level 10 m shuttle course at a speed controlled by an external audible bleep signal. The test progressively increases at 1 min interval for a total of 12 intervals and is terminated when the participant fails to complete a shuttle within the required time. Following the ISWT, a walking speed equating to 85% of predicted peak $O_2$ consumption ($VO_2$ peak) can be calculated, using a conversion table based on their ISWT performance. The ESWT is completed at this continuous speed on the same shuttle course until volitional exhaustion or the end of the test (20 min) is reached. For the purposes of this study, any participant who does not complete the full 20 min of the ESWT in visit one will be excluded to ensure standardisation of the test in visit 2. This protocol was initially developed for use in patients with chronic airways obstruction[37] but has been used in the CKD population,[39 40] showing good reproducibility and a high correlation with $VO_2$peak.[41]

### Venous blood sampling
Venous blood will be collected using venepuncture of the antecubital vein of either arm—provided the absence of an arteriovenous fistula. Blood will be drawn through a 21-gauge needle (30 mL per time point) and collected into di-potassium EDTA, sodium citrate, sodium heparin and blank monovettes.

### Blood processing, storage and analysis
Blood collected into EDTA tubes will be centrifuged at 4°C; the supernatant will then be aliquoted and frozen at −80°C for future analysis. A cytometric bead array technique will be used to allow bulk analysis of a panel of proinflammatory and anti-inflammatory cytokines, including but not limited to IL-1, IL-2, IL-6, IL-10, tumour necrosis factor alpha and interferon gamma. Blood collected into sodium citrate[42] will be double centrifuged at room temperature (15 min at 2500 $g$, supernatant aliquoted followed by another 15 min at 2500 $g$) to create platelet-poor plasma; the supernatant will then be aliquoted and frozen at −80°C for future phenotyping of microparticles using flow cytometry, as explained below. Blood collected into sodium heparin will be analysed on the day of collection for immune cell subsets using flow cytometry, as explained below. Blood collected into blank monovettes will be sent to the Diagnostic Pathology Service at University Hospitals of Leicester National Health Service (NHS) Trust on the day of collection for renal profile analysis, which includes estimated glomerular filtration rate, urea, bicarbonate, creatine, sodium, potassium and phosphate measures. This will be completed for both patient and control populations.

### Flow cytometry
Immune cells will be characterised based on their expression of surface antigens using flow cytometry. Monocyte subsets will be categorised as classical (CD14$^{++}$CD16$^-$), intermediate (CD14$^{++}$CD16$^+$) and non-classical (CD14$^+$CD16$^{++}$).[43] Monocyte angiotensin-converting enzyme expression will be assessed using CD143, with immunoglobulin G1 as a negative control. T cells will be categorised as helper (CD3$^+$CD4$^+$) cytotoxic (CD3$^+$CD8$^+$) and regulatory (CD4$^+$CD25$^+$CD127) T cells. B Cells will be identified as CD3$^+$CD19$^+$ and NK Cells as CD3-CD56$^+$. Following appropriate staining and washing procedures, immune cells will be analysed on an FACSCalibur (BD Biosciences, Oxford, UK). Acquisition templates and preliminary flow cytometry results are displayed in figure 2.

Microparticles will be characterised based on their size, Annexin-v expression, cellular derivation and prothrombotic potential. Cellular derivations will be categorised as platelet derived (CD42b$^+$), neutrophil derived (CD66b$^+$), monocyte derived (CD14$^+$) and endothelial cell derived (CD144$^+$). Prothrombotic potential will be estimated using Tissue Factor (CD142$^+$) expression. Following the thawing of platelet-free plasma at room temperature, samples will be double centrifuged at room temperature (30 min at 18 000 $g$, supernatant removed, discarded and replaced with an equal volume of buffer, followed by another 30 min at 18 000 $g$). Samples will then undergo appropriate staining procedures with the antibodies listed above, after which microparticles will be analysed on a BD Accuri C6 (BD Biosciences, Oxford, UK) flow cytometer. Acquisition templates and preliminary microparticle

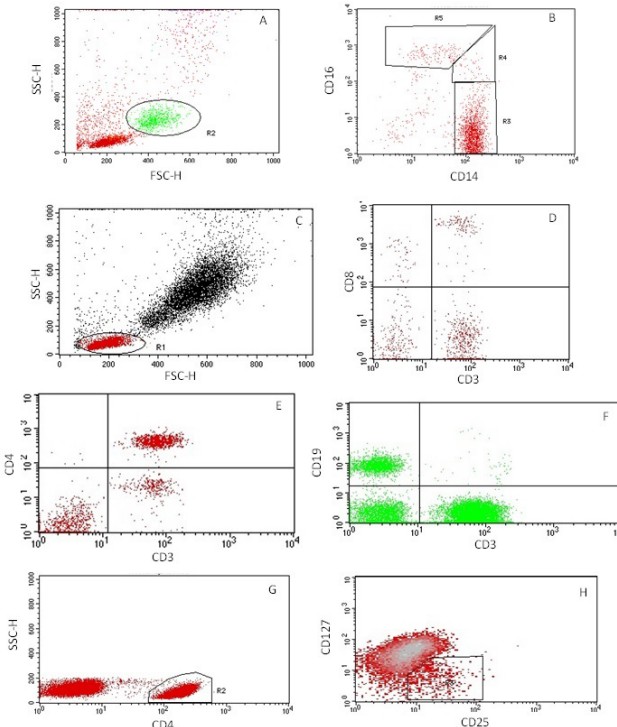

**Figure 2** Demonstration of the gating strategy used to identify immune cell subsets. (A) Total monocyte gating. (B) Monocyte subsets (R3 = classical (CD14++CD16-), R4 = intermediate (CD14++CD16+), R5 = non-classical (CD14+CD16++) (R1 and R2 were used as preliminary gating to remove neutrophils). (C) Total lymphocyte gating. (D) Gating CD8+ cytotoxic T lymphocytes in the upper right quadrant. (E) Gating CD4+ helper T lymphocytes in the upper right quadrant. (F) Gating B lymphocytes (CD3-CD19+) in the upper left quadrant. (G) Initial T-Reg gating, identifying CD4+ lymphocytes. (H) Secondary T-reg gating, back-gated onto plot G, further identifying the CD4+ lymphocytes that are CD25+CD127-. Not all graphs display 100% of acquired cells—this has been altered independently to allow ease of gating.

results are displayed in figure 3. The protocol for isolating and analysing MPs is based on a number of publications[42 44 45] and in our experience produces reliable results (figure 3).

### Saliva sampling and storage
Saliva samples will be collected into sterile plastic containers. Participants will swallow to empty the mouth, then open and hold the container themselves before performing a passive dribble of saliva collected under the tongue over the next 2 min. Following centrifugation, the supernatant will be aliquoted and frozen for future analysis, primarily for secretory IgA to investigate mucosal immunity.

### Anthropometric measures
Height, weight and waist and hip circumference will be measured using standard procedures.[46]

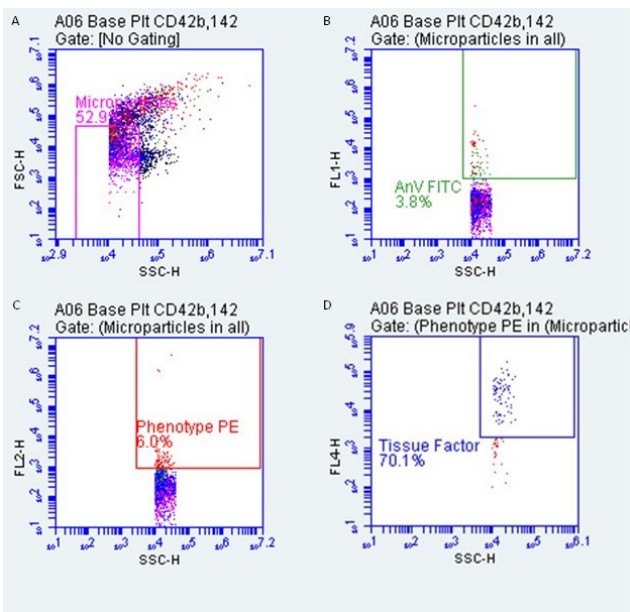

**Figure 3** Demonstration of the gating strategy used to characterise microparticles. (A) 'All microparticles' set using beads of known size. (B) 'All microparticles' based on Annexin-v expression. (C) Phenotype marker expression, used to quantify microparticles of different cellular sources. (D) Prothrombotic potential, assessed by Tissue Factor expression. All gates have been appropriately sized and positioned using unstained samples to distinguish negative versus positive staining. FITC, fluorescein isothiocyanate; F/SSC-H, Forward/Side Scatter-Height; PE phycoerythrin.

### Body composition
Body composition parameters (ie, fat and fat-free mass content) will be measured using bioelectrical impedance analysis (BIA) (Inbody 370, Chicago, Illinois, USA). BIA has been validated for use in CKD.[47] While BIA has limitations with regards to reliability and is not recommended for tracking changes in body composition, body composition will only be used as a descriptive rather than an outcome variable, and thus BIA has been deemed suitable to assess body composition for this reason.

### Cardiovascular condition and function
Cardiac bioreactance analysis will be carried out using a Non-Invasive Cardiac Output Monitor (Cheetah Medical, Maidenhead, UK). Cardiac bioreactance is a quick, safe, validated[48] and non-invasive method of assessing central haemodynamics including heart rate, stroke volume, cardiac output and total peripheral resistance. This is accomplished by using cutaneous electrodes placed on the chest in combination with an automatic sphygmomanometer to measure blood pressure.

### Time spent in leisure activities
The Leisure Time Exercise Questionnaire assesses the amount of exercise a patient undertakes, from which a metabolic equivalent estimation can be calculated. It also identifies preferred physical activities.

## Fatigue

Fatigue is a common complaint CKD, and it has multiple potential mechanisms including anaemia[49] and cachexia.[50] To assess self-reported fatigue, participants will be asked to fill in a fatigue scale (an internally designed Likert scale see online supplementary material), as well as the validated Functional Assessment of Chronic Illness Therapy-Fatigue[51] which has been extensively used in CKD populations[52]

## Muscle $O_2$ saturation

Reduced muscle $O_2$ saturation is another possible physiological mechanism of fatigue[53] and may be related to increased inflammation.[54] A small non-invasive muscle $O_2$ saturation device (BSXInsight, BSXAthletics, Texas, USA) will be fitted around the participant's calf and worn during exercise bouts and subsequent recovery. This device uses near infrared spectroscopy (NIRS) to measure haemoglobin and myoglobin oxygenation (muscle $O_2$ tissue stores). NIRS has previously been used in a variety of disease conditions.[55]

## Clinical information

Clinical records will be used to extract information that allows associations to be made with physical activity levels and account for confounding variables, including: age, gender, ethnicity, primary cause of renal failure, transplant or dialysis type, time since transplant or duration of dialysis, comorbidities and current medications.

## Data analysis plan

Mixed-design analyses of variance will be used to analyse main effects of group and time and interaction effects, with Bonferroni post hoc testing used to elucidate these effects. This will generate both tests of significance and estimates of effect sizes. Any non-normally distributed data will first be transformed appropriately to ensure normality assumptions are met, to increase statistical power and to allow clear conclusions to be drawn from the data. Where normality assumptions cannot be met via data transformation, Friedman's tests will be employed as a non-parametric alternative. This will allow the generation of hypotheses for future, larger studies.

## Ethics and dissemination

### Ethical and safety considerations

This protocol was reviewed by the East Midlands-Derby Research Ethics Committee (REC) and was given a favourable opinion (REC ref 15/EM/0391) on 24 September 2015. Additionally, local approval was given by the Research and Innovation office at the University Hospitals of Leicester NHS Trust (ref 11444) on 29 September 2015. Therefore, all steps have been taken when designing this protocol to minimise all ethical implications and ensure patient welfare. Two substantial amendments were made to this protocol, approved on 03 December 2015 and 01 August 2016 to add saliva collection and fatigue questionnaires and muscle $O_2$ saturation, respectively. The protocol presented here represents the most up-to-date version.

## Dissemination plan

The results are planned for publication in early 2018. We plan to present the data at relevant national and international conferences, as well as publish the findings in relevant journals. Participant level data will be available at a later date.

## Trial registration

This study is registered with the ISRCTN (ISRCTN38935454). The registration was completed after recruitment of the first patient, so the study was registered retrospectively.

**Contributors** PJH wrote this manuscript; JN was involved in study design and protocol preparation; TJW was involved in study design and preparation of protocol amendments; NCB was involved in study design; ACS lead study design and protocol preparation. All authors reviewed this manuscript.

**Funding** We are grateful to the Stoneygate Trust for partially funding this study. This work is supported by the National Institute for Health Research (NIHR) Diet, Lifestyle & Physical Activity Biomedical Research Unit based at University Hospitals of Leicester and Loughborough University. The views expressed are those of the authors and not necessarily those of the NHS, the NIHR or the Department of Health.

**Competing interests** None declared.

**Patient consent** Obtained from patients.

**Ethics approval** REC East Midlands-Derby (ref 15/EM/0391).

**Provenance and peer review** Not commissioned; externally peer reviewed.

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
