## [Reviewer comments · BMJ Open]

ARTICLE DETAILS

TITLE (PROVISIONAL)	Physical Activity, Immune Function and Inflammation in Kidney Patients (the PINK Study): A Feasibility Trial Protocol
AUTHORS	Highton, Patrick; Neale, Jill; Wilkinson, Thomas; Bishop, Nicolette; Smith, Alice

VERSION 1 - REVIEW

REVIEWER	Oleh Akchurin Weill Cornell Medicine, USA
REVIEW RETURNED	30-Oct-2016

GENERAL COMMENTS	In the manuscript entitled "Physical activity, Immune Function and Inflammation in Kidney Patients: the PINK Study Protocol", the authors propose to explore the acute changes in some of the immune system parameters following standardized aerobic exercise performed by patients with chronic kidney disease (CKD). The topic is timely and important, and the manuscript was reviewed with great interest. General comments: The manuscript is well written and clearly articulates authors' intent to investigate whether acute exercise has any measurable effect on the parameters of their interest in patients with CKD. While authors declared their project to be a "feasibility study", success of their research efforts may be further assured by additional clarifications of the protocol components. Even for the feasibility study, the authors may want to have a projected number of participants in mind. Otherwise it is not clear when the investigators will stop the recruitment and data acquisition, and initiate data analysis. Power analysis can be performed based on the similar studies conducted in general population, which the authors cite in the introduction (page 3, lines 8-10). Inclusion and exclusion criteria may deserve a second look. As written, inclusion criteria seem to be too broad, and as such patient heterogeneity may complicate interpretation of the results (e.g., stage 1 CKD patients would probably have quite different baseline immunity as compared with hemo- or peritoneal dialysis patients, as well as different exercise capability). Authors provided gating strategy for FACS studies of the peripheral blood samples, however other biomarkers of interest (i.e., cytokines, "renal profile", "phenotyping of microparticles") are not defined in the current version of the protocol. Data analysis plan is not clearly articulated either. Specific comments: Abstract: - It would be helpful to define the projected number of study participants and CKD stage(s) (page 2, line 13)
--

- Not clear what the authors mean by “standard assay” of inflammation and immune function; please elaborate both in the abstract and in the manuscript (page 2, lines 24-25).
- is 2017 realistic for projected publication (page 2, lines 30-31)? In the text of the manuscript the authors are also mentioning early 2018 (page 8, line 20). If patient recruitment is ongoing already, manuscript submission perhaps may be realistic for 2017-2018, but then the readers may be interested in knowing how many subjects have been recruited by the time of this manuscript submission.

Introduction:

- It may be helpful to at least briefly discuss the specific measures proposed in the protocol, such as cytokines, microvesicles, immune cell populations, and place them in the content of what is known already about these biomarkers with respect to exercise.
- While it is fair to use the data generated by the proposed study for power calculations to plan future larger studies, this does not mean that the power analysis cannot be performed for the current protocol (page 3, lines 30-31). One way or another, the authors will need to decide on the number of patients they want to enroll, and the readers would want to know that number and its justification.

Methods and Analysis:

- The source of patient recruitment is not clear, as the authors intend to screen “all CKD patients” (lines 36-37). All CKD patients seen by the nephrologists in their institution? Or in “the local community” by the primary care providers? What is the anticipated sample size of the patient population available for screening? What is the anticipated number of CKD patients to be identified through the screening? What is the anticipated percent of patients who will agree to participate among those who offered participation?
- Inclusion criteria, as written (page 4, lines 4-6), seem to be too broad. If the authors are planning to enroll patients with stage 1 CKD, this needs to be justified. It is not clear from Table 1 if the authors are planning to enroll dialysis patients. Only from the very end of the protocol (page 7, line 52) the readers may guess that dialysis patient indeed will be included. If this is truly the authors’ intent, they may want to define the temporal relationship between study visits and hemodialysis sessions, same for peritoneal dialysis treatments, and acknowledge somewhere that they will attempt to account for the effect of fluid overload on exercise and bioimpedance analysis. If feasible from the standpoint of eligible patient numbers, the authors may consider to focus on a rather narrow range of GFR, e.g., maybe just stage 4 CKD? Otherwise, the authors may face a problem of variability in outcomes, and inability to reliably adjust for GFR / CKD stage due to small or unequal numbers of patients with each CKD stage enrolled.
- Is there any upper age limit for study participation?
- Not clear what the authors mean by “own clinician” (page 3, line 43; page 4, lines 9-10). Primary care provider?
- What is the time frame between recruitment and visit 1, and between visit 1 and visit 2? The latter should probably be short to make sure there is no deconditioning between the visits, but this is not specified in the protocol. What are the anticipated drop-out rates between recruitment and visit one, visit one and visit two?
- How many pro-inflammatory cytokines will be measured (page 5, lines 21-22)? If one or two cytokines, then ELISA would be appropriate. If however the authors are planning to measure multiple cytokines, then they may want to use one of the multiplex techniques, if feasible. Anti-inflammatory cytokines (such as IL-10),

	may be also of interest. The readers would want to know what specific cytokines are proposed to be measured.  - Proposed analysis of the peripheral blood micro-vesicles is a potential strength of this study, but it is coming only at the end of the protocol (page 5, lines 26-27). What is the rationale for choosing citrate as an anticoagulant for micro-vesicle isolation? The proposed speed of the second centrifugation seems to be very low. The citation of the micro-vesicle isolation protocol that the authors intend to use would be helpful to readers. If using their own protocol, the authors may want to describe their experience with this technique. - What is “renal profile” (page 5, line 30)? Please consider clarification. Clinical information about study subjects obtained from their medical records (page 7, lines 48-54) will probably include serum creatinine and urea nitrogen for the CKD group? - Figure 2: Not clear how R4 was selected on the plot A. There is no designation of X-axis for the plot D – probably F11, and probably obscured by the superimposed plot F? It would be easier for the readers if antibody designation is shown on the figure, not only in the legend. - Please consider to clarify what blood and urine test results you are planning to abstract from medical records (page 7, lines 53-54). Will this be equally available for both CKD and control groups? - Data analysis plan (page 7, line 56 to page 8, line 5) needs further development. First sentence can be removed as it does not seem to provide much information to the readers. Second sentence outlines a quite complex design combining between- and within-group analyses without explaining how this will be executed. Analysis of interaction (page 8, lines 4-5) also needs to be elaborated. - Adding a section “Expected results, potential pitfalls and alternative solutions”, or similar may be helpful.
--	--

REVIEWER	Vincent James Dalbo Central Queensland University Australia
REVIEW RETURNED	15-Nov-2016

GENERAL COMMENTS	Reviewer Comments – Physical Activity, Immune Function and Inflammation in Kidney Patients: The PINK Study Protocol General Comments: I do not see the value in publishing this type of work as all of this information will appear in the published manuscript; however, I will accept with slight revisions and allow the editor to decide the value of the work. Overall, this was a well written document requiring additional information to be complete. I also feel strongly BIA should not be used to analyse body composition. Major comments: BIA should not be used to assess body composition. There is great variation in oxidative stress from individual to individual and day to day within the same individual. As a result, some researchers have questioned the applicability of examining markers of oxidative stress. I still believe the examination of
---

oxidative stress can provide value but understand the variability is why oxidative stress will ever be used in clinical practice. That said, ever step should be taken to help control for the variability in oxidative stress, such as testing during the same time of day (which is not mentioned in the manuscript), using men or women (which is not mentioned in the study), using patients at a similar stage of CKD (which does not occur), controlling for diet (which is somewhat controlled), and controlling for exercise. The biggest issue I have hear is using patients at all stages of CKD. From my knowledge in the area, someone with early stage CKD is very unlikely to benefit from walking while walking may prove to be enough of a stimulus to reduce oxidative stress in someone who has in the latter stages of CKD. It is very likely your results are going to be muddled by not controlling for stage of CKD.

Also, if publishing a methods paper it would be great to see a power calculation.

Title: I believe “the” should be capitalized.

Abstract: Line 11: Add “patients” following CKD.

What level of CKD patients will be in the study – it appears like all.

Introduction:

Interesting that the research suggests moderate aerobic exercise will potentially provide the best outcomes. Leading researchers in the field currently have investigations examining the effects of high intensity interval training (HIIT) in CKD patients and recent animal work by Tucker et al. suggests light intensity activity (LIA), which is a greater intensity than walking, has minimal effect on reducing oxidative stress and inflammation while HIIT was effective for reducing oxidative stress, enhancing oxidant defence mechanisms, reducing inflammation, and protecting kidney health in an animal model of early stage CKD.

I do not understand the rational for using walking as the treatment to CKD. I am familiar with the effects of walking and other forms of LIA and HIIT on health in healthy and diseased populations, specifically CKD and CVD, and research does not support the practicality of LIA to improve health in the vast majority of the populations – see Batacan et al. 2016. As a result, there should be some mention in the introduction in regard to the strengths of weaknesses of using a walking protocol to improve health in this population.

Methods:

Participant recruitment

I am confused by the wording with no know chronic disease. CKD is a chronic disease.

Venous blood sampling

	How much blood will be drawn at each time point? Body composition Body composition should not be obtained with BIA. The use of BIA is to measure body water and I can produced work that has found BIA to not be valid for the measurement of body composition. Please use an acceptable measure of body composition such as skin fold measures. See work by Moon et al. Data analysis plan No data analysis plan is provided the authors simply state the appropriate tests will be conducted instead of listing the tests that will be conducted for each variable. This needs to be amended.
--	--

REVIEWER	Jennifer Heaney University of Birmingham
REVIEW RETURNED	21-Nov-2016

GENERAL COMMENTS	The authors detail a protocol for a pilot investigation into the effects of acute exercise on immune function in kidney patients. The protocol is clear and well-written.  - In the intro the authors state that exercise recommendations in kidney patients are currently lacking. Is there any in place at the moment? If so if the authors could briefly touch upon this/provide a reference for any current guidelines to direct readers to. - For visit 2, all participants will arrive in the morning. Will this be during a controlled timeframe eg 8-9am, 8-10am? As immune parameters could potentially be influence by time of day, it may be important to ensure that both groups of participants perform the exercise bout within the same time window. - Caffeine, alcohol, exercise are controlled prior to the laboratory visit. Will participants be fasted? Or be advised on breakfast and their timing? Guidelines regarding eating/drink on the morning of the visit may be helpful to ensure standardisation between groups. - Can more information regarding how the VO2 peak will be provided? Is this just based on the stage of the test (please describe) or will expired air be collected/is HR measured? Will HR be monitored during the test – this may be useful and will provided information regarding the % predicted max HR achieved during the tests and between groups. - How will a constant walking speed be maintained and enforced during the ESWT? Is a shuttle walk test the most appropriate test for physical performance for the 2nd visit – the participant will be walking up and down the same 10m (presumably requiring 180 degree turns..maybe a causing a pause in the walking) for 20 mins. Is there a possibility that participants may drop out due to boredom rather than exhaustion. If testing is being conducted in a laboratory, would a treadmill, set to the appropriate speed determined from test
---

	1, be appropriate?  - The shuttle tests were initially developed for use in pulmonary patients. Have they been used/validated in this population with renal disease previously? - The positive effects of exercise seen previously cited in refs 26-29 relate to exercise training. Even if no favourable changes on immunity and inflammation are seen in this acute study, aerobic exercise training should still be investigated in these patients in relation to these outcome measures.
--	---

VERSION 1 – AUTHOR RESPONSE

Reviewer 1

The number of projected participants has now been included (through the use of a power calculation) in the introduction (page 3, paragraph 4) however it has not been included in the abstract in order to keep it as brief as possible. This calculation was based on the lymphocyte data from the acute exercise section of the study by Viana et al (2014 – reference number 34 in the manuscript). This was completed using GPower 3.1 software, and was based on a power of 0.8 and an alpha level of 0.05. As the study only involves 2 brief visits, this does not include a dropout rate.

'Standard assay' has now been updated with more information, both in the abstract and in the main body ('Blood processing, storage and analysis', page 5).

The proposed timeline for publication has been adjusted to publication in 2018, both in the abstract and in the main body ('Dissemination Plan', page 10).

More information regarding the specific measures (cytokines, microparticles etc.) has been entered in to the introduction (page 3, paragraphs 1 and 2).

A power calculation has now been completed ('Participant recruitment', page 3). As this protocol is designed to allow us to potentially investigate a number of different populations (i.e. pre-dialysis CKD patients, dialysis patients, renal transplant recipients and healthy controls) the necessary number of participants per group has been calculated, rather than the total number.

More information has been entered with regards to the source of patient recruitment ('Participant recruitment', page 3). Estimating the total number of available patients is difficult, however as our power calculation suggests that 15 participants per patient population will be sufficient, we expect that the pool of available and subsequently eligible patients within the Leicester renal network will be more than large enough to allow successful recruitment.

Information regarding the broadness of the inclusion criteria has been added to the 'Participant recruitment' section (page 3). Dialysis patients will be eligible (now added to the inclusion/exclusion criteria, Table 1), and we have added more information regarding the timings of the visits with regards to haemodialysis patients (page 4, paragraph 4).

There is no upper age limit for participation (exclusion criteria only states under 18 years of age).

However, as exclusion criterion is 'Any element of study assessment protocol considered by primary care provider to be contraindicated due to physical impairment, comorbidity or any other reason' it is unlikely that any extremely elderly participants will be enrolled in the study.

'Own 'clinician' has been updated to 'principle care provider' ('Participant recruitment', page 3 and Inclusion/Exclusion Criteria, Table 1 page 4). This is intended to represent the individual who is responsible for managing the treatment of the patient (e.g. a Consultant Nephrologist).

More information regarding the time-frame for recruitment and visits 1 and 2 has been added (page 4, paragraph 4).

Some proposed pro-inflammatory cytokines have been added ('Blood processing, storage and analysis', page 5). This is not designed to be a definitive list of the cytokines measured, but rather a flexible list that may be adjusted if necessary (e.g. in reaction to other subsequent publications etc.). Additionally, due to the large number of proposed cytokines (including anti-inflammatory cytokines)

we will implement a cytometric bead array technique, therefore allowing bulk-analysis.

The rationale for using sodium citrate is based around the paper by van der Heyde et al (now included in the manuscript – reference 42) which suggests that sodium citrate is the optimal anticoagulant for use in collecting venous blood for the purposes of microparticle analysis. Similarly, whilst the second centrifugation speed is fairly low, this is purely designed to create platelet-poor plasma for storage.

After thawing, samples are centrifuged again at 18,000g for a total of 60 minutes (now included in the manuscript – page 6, paragraph 2). Preliminary use of this protocol has shown it to be successful with regards to microparticle isolation. Our proposed gating strategy is also now included in Figure 3.

The measures included in the 'renal profile' assessment have been added ('Blood processing, storage and analysis', page 5). This is designed to characterise the chronic kidney disease population, and distinguish them from the healthy control population.

Figure 2 has now been updated to elucidate gating strategies, including antibody designation now shown on the figure.

The blood test results planned to be extracted from clinical records where those of the 'renal profile' as this is how we obtain the results from the pathology lab. This has been removed from the 'Clinical information' section in order to prevent confusion.

More specific information regarding the data analysis plan has been added ('Data analysis plan', page 9).

Reviewer 2

We are slightly confused by your comments regarding the measurement of oxidative stress. We have not mentioned oxidative stress in the manuscript, nor do we plan to measure it in any way. However, some of your points do also apply to measures of inflammation and immune function. We hope that either the changes or comments made in the manuscript regarding the time of day of testing (page 4 paragraph 3), the stages of CKD recruited ('Participant Recruitment', pages 3 and 4), and diet and exercise (page 4, paragraph 2). As is now mentioned in the manuscript, the inclusion of a wider variety of CKD stage will allow for the investigation of the influence of level of renal function with regards to the measures of immune function and inflammation proposed.

'Patients' has been added in the abstract.

With regards to your comments concerning the intensity of the walking exercise; you are correct in saying that high-intensity interval training may confer a greater health advantage than lighter intensity exercise. However, this study is designed to investigate a pragmatic form of exercise that CKD patients could feasibly complete independently. As the goal is to ultimately inform exercise recommendations for the entire CKD patient population, it seems more pertinent to investigate a 'more attainable' form of exercise. From our experience, many CKD patients have very little exercise experience and would find High-Intensity Interval Training (HIIT) either too intimidating initially, or they may even be unsuitable for HIIT due to their high prevalence of cardiovascular disease (HIIT in CKD patients is currently under-researched). With that being said, the ISWT and ESWT protocol is designed to elicit walking at a pace equivalent to 85% VO₂peak, and thus would be considered moderate (if not high intensity) rather than light intensity exercise. Whilst the goal of this study is not to improve the health of the CKD population but rather simply investigate the effects of acute aerobic exercise on immune function and inflammation, previous research has displayed that regular moderate intensity walking exercise can elicit meaningful improvements in inflammation (i.e. reduction in the ratio of plasma IL-6 to IL-10 levels) and immune function (down-regulation of T lymphocyte and monocyte activation) in CKD patients (see Viana et al, J Am Soc Nephrol 25: 2121–2130, 2014). The wording 'no known chronic disease' relates to the healthy control group, as mentioned earlier in the sentence (page 3, paragraph 6).

The volume of blood collected at each time-point has been included (page 5, paragraph 2).

Whilst your concerns regarding BIA are valid, in this case body composition is being used as a characterisation measure in order to comment on any differences/similarities present between the groups at baseline as such is not considered as an 'outcome measure' per se. Skin-fold measures require significant investigator experience and as such can be subject to inter-investigator variation,

increasing the logistical burden of the study as well as the burden on the participant (i.e. requirement of removing clothing etc.)

More information regarding the proposed statistical tests has been included (page 9, paragraph 7).

Reviewer 3

A reference regarding exercise recommendations in CKD (Johansen, 2005) has been added (page 3, paragraph 4), with a comment concerning the severe lack of information regarding immune function and inflammation.

The time-frame for visit 2 has been added (page 4), as you are correct in saying that diurnal variation may impact on certain measures and thus must be at least partially controlled for.

Participants will not be fasted (added in page 4) – whilst this may prevent a certain amount of standardisation, as this is a pilot study every effort has been made to reduce the burden on the participant, and we deemed controlling for caffeine, alcohol and exercise to be more important with regards to immune function and inflammation.

Regarding the points raised concerning the ESWT: The VO₂peak is estimated using a conversion table based upon the ISWT performance (information added to page 5, paragraph 1). Whilst expired air collection would allow for a more precise measure of VO₂peak, this method has been shown to be valid (see Painter 2012) and is far simpler with regards to equipment requirements and participant burden.

A constant walking speed is maintained during the ESWT by both the adherence of the participant to the auditory beeps on the CD and encouragement by the investigator. The protocol for this test dictates that 2 consecutive missed beeps (i.e. participant 50cm short of the cone when the beep sounds) is grounds for termination of the test. Therefore, if a participant begins to slow down and misses one beep, they are encouraged to speed up. If they fail to do so, the test is ended and the participant will be withdrawn from the study due to their inability to complete the intervention.

Whilst the participants do complete a 180 degree turn at the end of every shuttle, they usually set into a good rhythm and turn without a pause and therefore maintain the same level of intensity throughout the 20 minutes.

From our experience, although participants have complained of boredom it is usually not severe enough to cause a dropout as it is only 20 minutes long.

Lastly, although a treadmill would be useful for ensuring the appropriate speed is maintained, there are 2 main reasons why we have chosen to use the combination of the ISWT and ESWT. Firstly, it is from our experience in other studies in CKD patients that a large proportion of them are uncomfortable using treadmills due to balance issues etc. Similarly, in order to standardise the 20 minutes we would first need to do a treadmill-based peak test, which some patients may find intimidating and would therefore reduce recruitment success. The ISWT/ESWT system provides an easy, non-threatening and widely-used (see addition to protocol) way to achieve effectively the same thing. It is also more pragmatic, as if a patient would decide to increase their walking activity they would be more likely to 'go for a walk' than use a treadmill. Secondly, as this is a pilot study, every effort has been made to simplify where possible and thus the ISWT/ESWT is more advantageous with regards to equipment and space.

VERSION 2 – REVIEW

REVIEWER	Oleh Akchurin USA
REVIEW RETURNED	22-Jan-2017

GENERAL COMMENTS	The quality of the protocol / article has much improved. It would help to specify in your new power calculations section what power was used for calculations of the projected sample size (e.g., 0.80?) and what alpha level (0.05?). On page 7 you are planning to transform all
--

	non-normally distributed variables to normally distributed, but this is not always possible. You can, however, use statistical tests that do not require normal distribution.
--	---

REVIEWER	Vincent Dalbo Central Queensland University Australia
REVIEW RETURNED	15-Jan-2017

GENERAL COMMENTS	Thank you for responding to my requests. I have two minor issues remaining. 1. This information should be included with your power calculation: on a power of 0.8 and an alpha level of 0.05. Also, please state which variable from the Viana et al. manuscript was used to derive the sample size of 15. 2. Whilst your concerns regarding BIA are valid, in this case body composition is being used as a characterisation measure in order to comment on any differences/similarities present between the groups at baseline as such is not considered as an 'outcome measure' per se. Given this is a methods paper I can see future investigations that are seeking to track body composition citing your work for justification of using BIA to track body composition. Please acknowledge in your manuscript the limitations of BIA and state BIA was used only because body composition was used as a descriptive rather than an outcome variable.
---

REVIEWER	Jennifer Heaney University of Birmingham, UK
REVIEW RETURNED	24-Jan-2017

GENERAL COMMENTS	The authors have adequately addressed my previous comments and questions.
---

VERSION 2 – AUTHOR RESPONSE

Reviewer 1

The information concerning the power calculation, including the power and alpha level and the data it is based on has now been entered into the manuscript (page 3, paragraph 4).

We have included a non-parametric alternative data analysis plan for data that cannot be appropriately transformed ('Data Analysis Plan', page 7, paragraph 2).

Reviewer 2

The information concerning the power calculation, including the power and alpha level and the data it is based on has now been entered into the manuscript (page 3, paragraph 4).

We have included a short section acknowledging the limitations of BIA with regards to tracking body composition, and have highlighted that it is only being used for the purposes of a descriptive variable in this case ('Body Composition', page 6, paragraph 3).

VERSION 3 – REVIEW

REVIEWER	Oleh Akchurin Weil Cornell Medicine, USA
REVIEW RETURNED	10-Feb-2017

GENERAL COMMENTS	The quality of the protocol / manuscript keeps improving. Few more suggestions: Abstract - Methods and Analysis: last sentence of this paragraph: please be more specific about your plan for data analysis. At the very least mention the specific test for which you powered the study. Page 3, lines 35-36: should be "lymphocyte count", not "lymphocyte concentration" Page 4, exclusion criteria: I wouldn't rely on primary care providers in evaluating contraindications for study participation, first, because I am assuming they will not be your official co-investigators on this study, and second, they may not have enough expertise to perform such evaluation, as some of the study procedures are out of general practitioners' scope of practice. Page 5, Flow Cytometry: disabbreviate ACE Page 6, Fatigue: please submit your internally designed fatigue scale as supplementary material. What are its advantages over validated FACIT-F? Page 7, Data analysis plan. The last clause of first sentence is still too vague. You can not say that you will "investigate any other pertinent outcomes". Please be specific. Lymphocyte count seems to be your primary outcome, since you powered your study for it? Other cell counts, beads and cytokines are secondary outcomes. Figure 2: please move R3 designation out of the middle of cell population to the periphery of the box on panel B (on the current version it is obscured and difficult to read). Explain in the legend what R1 and R2 stand for. Your axes legends partially obscure values under axes (e.g., Panel B). There is some text partially visible on the top of panel G: either remove completely or show fully if it is pertinent.
--

VERSION 3 – AUTHOR RESPONSE

Reviewer 1

The complete plan for the data analysis cannot be included in the abstract as it would become overlong and complex. In order to maintain appropriate brevity of the abstract, we have included the necessary information in the main body of the text (page 7, 'Data Analysis Plan').

Page 3, lines 35-36: 'lymphocyte concentration' has been changed to 'lymphocyte count'

Page 4: exclusion/inclusion criteria. While we appreciate the reviewer's point here, we must stress that UK NHS ethics regulations require the patient's own clinician to assess their suitability for the study and we are therefore following national guidelines in this respect. This study will be conducted in a large renal referral setting which is very research active. All the renal clinicians who will screen and refer patients to the study work closely with the research team and are well aware of the study protocol and research procedures: they are not general practitioners but specialist renal consultants.

To clarify, we have altered the wording on page 3 (Participant Recruitment) to “consultant nephrologist” rather than “principal care provider”.

Page 5, Flow Cytometry: ‘ACE’ has been changed to ‘Angiotensin-Converting-Enzyme

Page 6, Fatigue: Our internally designed Fatigue Visual Analogue Scale has been included in the submission as Supplementary Information as requested. We chose to use this as well as the FACIT-F to provide a preliminary exploration of the patient perception of different “fatigue” experiences i.e. muscular, respiratory and mental fatigue. This approach was initiated by informal discussions with our patient population who indicated that the concept varied between individuals. The information provided by this VAS is intended to inform future research into this symptom which is very common in kidney patients.

Page 7, Data Analysis Plan: The first sentence in this paragraph has been adjusted. The aim of this sentence is to highlight that we will statistically analyse both the differences between groups at baseline and the effects of the exercise. Some variables (such as the questionnaire and cardiovascular function data) are only measured at baseline and as such can only be compared between groups at this time-point, whilst other variables (such as the cell counts, microparticles and cytokine) are measured at all three time-points and thus the effect of exercise can be assessed in these variables.

Figure 2: the R3 designation has been moved so it is more clearly visible. The meaning of R1 and R2 has been explained in the figure caption. The figure has been adjusted so that no axes values are obscured and no unnecessary text is visible.

VERSION 4 – REVIEW

REVIEWER	Oleh Akchurin Weill Cornell Medicine, USA
REVIEW RETURNED	28-Mar-2017

GENERAL COMMENTS	Thank you for making changes. Good luck with the study!
---